# Structure of the Calvin-Benson-Bassham sedoheptulose-1,7-bisphosphatase from the model microalga *Chlamydomonas reinhardtii*

**Théo Le Moigne[1,2], Martina Santoni[1,3], Lucile Jomat[1], Stéphane D Lemaire[1], Mirko Zaffagnini[3], Nicolas Chéron[4], Julien Henri[1]\***

[1]Sorbonne Université, CNRS, Laboratoire de Biologie Computationnelle et Quantitative UMR 7238, Institut de Biologie Paris-Seine, Paris, France; [2]Faculty of Sciences, Doctoral School of Plant Sciences, Université Paris-Saclay, Saint-Aubin, France; [3]Department of Pharmacy and Biotechnology, University of Bologna, Bologna, Italy; [4]PASTEUR, Département de chimie, École Normale Supérieure (ENS), PSL University, Sorbonne Université, CNRS UMR8640, Paris, France

**\*For correspondence:**
julien.henri@sorbonne-universite.fr

**Competing interest:** The authors declare that no competing interests exist.

## eLife Assessment

This study provides a **valuable** structural analysis of the Sedoheptulose-1,7-Bisphosphatase (SBPase) from *Chlamdomonas reinhardtii*. The data presented are **solid** and based on X-ray structures of the CrSBPase in an oxidized and reduced state, the authors identify a disulfide bond in close proximity to the dimer interface. They show that the redox-state of the CrSBPase impacts its oligomeric state and might also influence the activity of the protein.

**Abstract** The Calvin-Benson-Bassham cycle (CBBC) performs carbon fixation in photosynthetic organisms. Among the eleven enzymes that participate in the pathway, sedoheptulose-1,7-bisphosphatase (SBPase) is expressed in photo-autotrophs and catalyzes the hydrolysis of sedoheptulose-1,7-bisphosphate (SBP) to sedoheptulose-7-phosphate (S7P). SBPase, along with nine other enzymes in the CBBC, contributes to the regeneration of ribulose-1,5-bisphosphate, the carbon-fixing co-substrate used by ribulose-1,5-bisphosphate carboxylase/oxygenase (Rubisco). The metabolic role of SBPase is restricted to the CBBC, and a recent study revealed that the three-dimensional structure of SBPase from the moss *Physcomitrium patens* was found to be similar to that of fructose-1,6-bisphosphatase (FBPase), an enzyme involved in both CBBC and neoglucogenesis. In this study we report the first structure of an SBPase from a chlorophyte, the model unicellular green microalga *Chlamydomonas reinhardtii*. By combining experimental and computational structural analyses, we describe the topology, conformations, and quaternary structure of *Chlamydomonas reinhardtii* SBPase (*Cr*SBPase). We identify active site residues and locate sites of redox- and phospho-post-translational modifications that contribute to enzymatic functions. Finally, we observe that *Cr*SBPase adopts distinct oligomeric states that may dynamically contribute to the control of its activity.

## Introduction

Photosynthetic carbon fixation is performed by eleven enzymes that collectively operate the CBBC (*Calvin, 1962*; *Johnson, 2016*), wherein Rubisco plays the prominent role in catalyzing fixation of inorganic carbon onto a pentose acceptor: ribulose-1,5-bisphosphate (RuBP). The net gain of the CBBC is one triose in the form of glyceraldehyde-3-phosphate (G3P) for every three carboxylation reactions catalyzed by Rubisco (*Johnson and Alric, 2013*). Regeneration of RuBP is ensured by the other ten enzymes of the CBBC, among which phosphoribulokinase (PRK) and SBPase are the only enzymes that are exclusively involved in this pathway. The remaining eight enzymes are paralogs of those involved in cytoplasmic glycolysis, gluconeogenesis, or the pentose-phosphate pathway. Hence, Rubisco, PRK, and SBPase are expressed in photosynthetic organisms and fulfill biochemical functions so far uniquely attributed to the photosynthetic carbon fixation.

Pioneering studies on SBPase were carried out using the native enzyme purified from wheat chloroplasts (*Woodrow, 1982*) and corn leaves (*Nishizawa and Buchanan, 1981*), allowing quantitative assays of its catalytic function. This enzyme catalyzes the hydrolysis of SBP into sedoheptulose-7-phosphate (S7P) and requires magnesium ions for both catalysis and activation. In addition, SBPase activity is modulated by pH and redox switching mechanisms (*Anderson, 1974*; *Woodrow and Walker, 1980*; *Cadet and Meunier, 1988*). Redox control is exerted through dithiol/disulfide interchanges dependent upon the ferredoxin-thioredoxin cascade arising from the activity of the illuminated photosystems (*Schürmann and Buchanan, 1975*). In tobacco leaf extracts, SBPase reaches its maximal activity after 10 min illumination (*Zimmer et al., 2021*), and this activation kinetics is consistent with the slow reduction pattern of *Arabidopsis* SBPase in vitro (*Yoshida and Hisabori, 2018*).

The crystal structure of SBPase from the moss *Physcomitrium patens* (*Pp*SBPase) has been solved at a resolution of 1.3 Å (Protein Data Bank entry: 5iz3) (*Gütle et al., 2016*). *Pp*SBPase fold is highly similar to that of moss FBPase, yet the protein was shown to distinctively oligomerize as a homodimer and to form a disulfide bridge in a different motif than that found in plant FBPase (*Chiadmi et al., 1999*; *Gütle et al., 2016*). In spite of their homology, the sequences of FBPase and SBPase seem to have evolved different regulatory and catalytic properties that allow different contributions to the CBBC. Indeed, both enzymes specifically recognize their respective 6-carbon and 7-carbon sugar substrates (i.e. fructose-1,6-bisphosphate and SBP, respectively), with only SBPase capable to hydrolyze the other substrate with a resulting catalysis lower than that measured with SBP (*Gütle et al., 2016*). To date, the structural components contributing to active site selectivity (i.e. substrate recognition) can be retrieved from two studies reporting the crystal structure of the dual function fructose-1,6-bisphosphatase/sedoheptulose-1,7-bisphosphatase (F/SBPase) from *Thermosynechococcus elongatus* in complex with SBP (*Cotton et al., 2015*) and from *Synechocystis* sp. PCC 6803 in complex with FBP (PDB entry 3RPL). Recently, it was proposed that the CBBC can bypass fructose-1,6-bisphosphate (FBP) altogether, by only relying on SBP, glycolysis, and the oxidative pentose-phosphate pathway paralogs to regenerate the substrate of Rubisco (*Ohta, 2022*).

Several studies have reported that SBPase, along with Rubisco, FBPase, aldolase, and transketolase, constitute a metabolic bottleneck for the CBBC (*Raines, 2022* and references therein). Overexpression of SBPase was found to effectively enhance photosynthesis and growth in plants (*Lefebvre et al., 2005*; *Feng et al., 2007a*; *Feng et al., 2007b*; *Rosenthal et al., 2011*; *Simkin et al., 2017*) and in the green microalga *Chlamydomonas reinhardtii* (*Hammel et al., 2020*), indicating that SBPase can be considered as a limiting factor for the photosynthetic carbon fixation. Based on these evidences, it is plausible to consider that SBPase is a good target to improve the photosynthetic capacity by means of metabolic engineering coupled to synthetic biology (*Kubis and Bar-Even, 2019*), especially in the experimentally convenient multi-omic model alga Chlamydomonas (*Mettler et al., 2014*; *Schmollinger et al., 2014*).

In Chlamydomonas, SBPase is encoded by a single nuclear gene (Cre03.g185550), and the protein (Uniprot entry P46284) is addressed to the chloroplast by a 54-residue amino-terminal peptide (*Emanuelsson et al., 2007*; *Almagro Armenteros et al., 2019*). SBPase abundance in the algal cell was quantified by mass spectrometry as 1.1–1.3±0.3 amol/cell, representing 0.15% of total cell proteins (*Hammel et al., 2020*). In Chlamydomonas protein extracts, SBPase was shown to undergo post-translational modifications by phosphorylation of serines and threonines (*Werth et al., 2017*), and redox modifications of cysteine thiols (*Zaffagnini et al., 2012*; *Morisse et al., 2014*; *Pérez-Pérez et al., 2017*). However, the mechanisms by which post-translational modifications affect the catalytic

activity of CrSBPase is still lacking. Here, we describe the crystal structures of *Cr*SBPase in two redox states, examine the catalytic activity of recombinant CrSBPase, and investigate the correlation between the redox state and the oligomerization of the protein. Our results suggest that a redox control over oligomer equilibrium could represent a novel mechanism to regulate enzymatic activities in the CBBC.

## Results

### Crystal structure of *Cr*SBPase

Pure recombinant *Cr*SBPase crystallized in space group P2₁2₁2 and provided a complete X-ray diffraction dataset at a resolution of 3.09 Å. The asymmetric unit is composed of six polypeptide chains packed as three dimers and four water molecules could be modeled. All chains align within a root mean square deviation (RMSD) <0.25 Å and can be considered identical. 309 out of the 331 residues were built in the electron density, the unmodeled sequences are in the disordered extensions at the amino- and carboxy-termini of the model. *Cr*SBPase folds into two distinct domains (*Figure 1A–B*). The amino-terminal domain (NTD, residues 73–247) is composed of helices 1, 2, and 3 and a mixed β-sheet formed by strands 3–9. The carboxy-terminal domain (CTD, residues 248–383) is composed of helices 4, 5, and 8 and helices 6 and 7 that sandwich a mixed β-sheet formed by strands 10–13. α-helices 1–3 pack on the amino-terminal end of the seven-strand sheet, while α-helices 4,5, and 8 pack on the carboxy-terminal end of the four-strand sheet. α-helices 6 and 7 are sandwiched in between the two sheets. *Cr*SBPase chain A aligns with previously reported crystal structure of *Pp*SBPase with an RMSD of 0.478 Å (*Gütle et al., 2016* PDB 5iz3). This close structural similarity between orthologues indicates common structural properties supporting a conservation of the SBPase function in the evolutionary time separating the microalga Chlamydomonas from the land moss *P. patens*. A short antiparallel β-sheet is constituted of 4-residue strands 1 and 2 separated by a β-turn and preceded by loop A₁₁₃SCAGTAC₁₂₀. This motif projects to solvent from α-helices 2 and 3 (*Figure 1C–D*). Residues 113–130 forming the loop and β-hairpin (LBH) motif are positioned away from the core of the protein, at 15.3 Å from water molecule 401 (W401), and will be further discussed in the next sections.

AF2 entry AF-P46284-F1-model_v4 from AlphaFold Protein Structure Database aligns with our crystal structure 7b2o chain E with RMSD = 0.434 Å, showing excellent agreement between experiment and prediction at the level of protein main chain. Exceptions are in local differences in several loops conformations and in the N-ter and C-ter borders of secondary structure elements. Numerous amino acid residues side chains adopt moderately distinct orientations between the computational model and the experimental structure.

AF3 was recently communicated (*Abramson et al., 2024*) along with its online prediction server hosted at https://golgi.sandbox.google.com/. *Cr*SBPase model from AF3 align to our crystal structure 7B2O chain A with RMSD = 0.441 Å showing again their strong similarity and with a small discrepancy between AF2 and AF3 models with RMSD = 0.247 Å. The only significant deviations between 7B2O and AF3 are in the orientation of several side chains and notably on the conformation of region 114–131 that contain the LBH motif.

### CrSBPase catalytic pocket

*Cr*SBPase is structurally similar to its ortholog from the moss *Physcomitrium patens* (RMSD = 0.478 Å) (*Gütle et al., 2016*). SBPase is a homolog of FBPase, another enzyme involved in the CBBC in photosynthetic organisms and also ubiquitously involved in neoglucogenesis. Cytosolic isoforms of FBPase from non-plant sources have been extensively studied (*Moorhead et al., 1994*; *Lee and Hahn, 2003*). For instance, the crystal structure of human muscle FBPase complexed with its substrate (FBP) was solved (PDB entry: 5l0a, to be published). We aligned the structure *Cr*SBPase onto this human FBPase:FBP complex from entry 5l0a, confirming their similarity (RMSD = 0.824 Å) with the notable exception of the LBH motif which is absent in FBPase. In a 4 Å distance to FBPase co-crystallized with FBP we aligned 12 residues that are likely available for FBP or SBP binding by *Cr*SBPase (*Figure 1E*): E155, D173, D176, G177, Y287, G289, G290, M291, K317, L318, R319, and E323. These residues are exposed to the solvent in a continuous surface (*Figure 1F*). W401 could be modeled at the center of the putative SBPase catalytic pocket, in the vicinity of E155, D173, D176, R319, and E323, in the catalytic pocket predicted from alignment with human FBPase bound to FBP. W401 specifically is

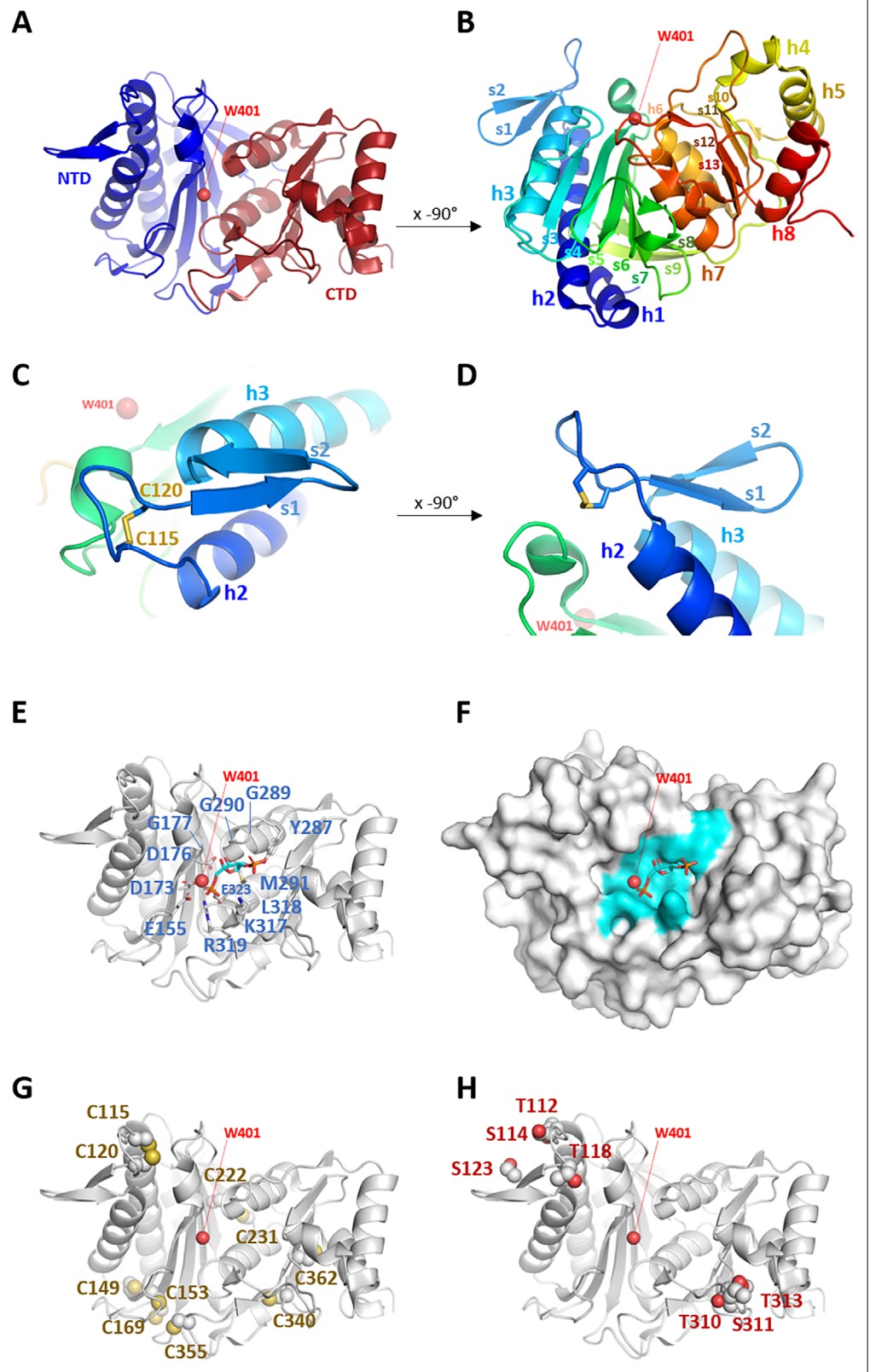

**Figure 1.** Crystal structure of *Chlamydomonas reinhardtii* SBPase (*Cr*SBPase). (**A**) Topology is displayed as cartoon with main chain colored from blue (amino-terminus) to red (carboxy-terminus). (**B**) Rotated view of A. by x-axis over –90°. (**C**) Close-up view of region A$_{113}$SCAGTAC$_{120}$ with disulfide bond shown in sticks. (**D**) Rotated view of C. by x-axis over –90°. (**E**) Putative active site residues inferred from alignment with fructose-1,6-bisphosphatase (FBPase)

*Figure 1 continued on next page*

*Figure 1 continued*

bound to fructose-1,6-bisphosphate (FBP) (5l0a, human muscle fructose-1,6-bisphosphatase E69Q mutant in active R-state in complex with fructose-1,6-bisphosphate) are represented in sticks. Residue numbering is according to Chlamydomonas sedoheptulose-1,7-bisphosphatase (SBPase) Uniprot entry P46284. (**F**) Surface representation in the B. orientation with putative active site residues colored in cyan. Water molecule 401 (W401) oxygen is represented as a red sphere. (**G**) Cysteines site chains are represented in spheres. (**H**) Threonines and serines side chains reported to be the target of phosphorylations are represented in spheres.

The online version of this article includes the following source data and figure supplement(s) for figure 1:

**Figure supplement 1.** Purification of recombinant *Chlamydomonas reinhardtii* SBPase (*Cr*SBPase).

**Figure supplement 1—source data 1.** Original gel image shown in *Figure 1—figure supplement 1B*.

**Figure supplement 1—source data 2.** Original gel image shown in *Figure 1—figure supplement 1B* (labelled).

**Figure supplement 2.** Cysteines pair C222-C231 close-up view.

**Figure supplement 3.** Sedoheptulose-1,7-bisphosphatase (SBPase) multiple sequences alignment.

in bonding distance to catalytic residues D178, D181, and E328. W401 is likely to hydrolyze the substrate SBP.

## Sites of redox post-translational modifications

Ten cysteines could be located in each monomer of *Cr*SBPase (C115, C120, C149, C153, C169, C222, C231, C340, C355, C362). Previous redox-based proteomic studies exploiting Chlamydomonas cells or protein extract established that *Cr*SBPase undergoes redox modifications by thioredoxin-dependent dithiol/disulfide exchange (*Lemaire et al., 2004*; *Pérez-Pérez et al., 2017*),S-glutathionylation (*Zaffagnini et al., 2012*), or S-nitrosylation (*Morisse et al., 2014*). While the putative Cys sites for S-glutathionylation are not known, thioredoxin and S-nitrosylation Cys targets were identified as being C222/C231/C355/C362, and C355/C362, respectively. DTNB-based thiol titration on the native purified SBPase indicates that 4.0±0.6 cysteines are likely reactive owing to their relative exposure to solvent in a free thiol form. Amino acid residue accessible surface area and accessibility (ASA) calculation is consistent with DTNB titration, with C149, C153, C231, and C355 exposed to the solvent (ASA respective areas: 0.154 Å², 0.113 Å², 0.308 Å², and 10.083 Å²), and the six others either shielded from solvent (C169, C222, C340, and C362) or engaged in a disulfide bridge (C115, C120) (*Ahmad et al., 2004*). In *Cr*SBPase crystal structure, C222 and C231 are positioned on neighboring strands of the amino-terminal domain, exposing their thiol 3.5 Å away from each other (*Figure 1G*, *Figure 1—figure supplement 2A–B*). The formation of an intramolecular disulfide bridge at this cysteines pair is possible if local main chain rearrangements allow a closer contact. C362 thiol is positioned 10.5 Å away from the nearest thiol group of C340. Additionally, C340 and C362 are exposed on an opposite side of the carboxy-terminal β-sheet, disfavoring the formation of an internal disulfide bridge. C355 points its side chain towards C153 and C169 with a thiol-thiol distance of 4.6 Å and 10.1 Å, respectively. Although not observed in our model, a C355-C153/C169 dithiol-disulfide exchange is thus plausible, as described for plant FBPase (*Jacquot et al., 1997*). Local rearrangements that occur upon FBPase disulfide reduction causes a larger scale relaxation of the active site required to reach maximal catalytic potential (*Chiadmi et al., 1999*) as kinetically characterized for enzymes from *Arabidopsis thaliana* (*Yoshida and Hisabori, 2018*) and *Nicotiana benthamiana* (*Zimmer et al., 2021*). In Chlamydomonas and Physcomitrium SBPases, the FBPase-type regulatory disulfide is not observed but the SBPase-type C115-C120 bridge is conserved in a nearly identical position in the LBH motif where it may regulate the enzyme by alternative conformational rearrangements. Indeed, the disulfide-bonded cysteines constrain Chlamydomonas SBPase loop $A_{113}SCAGTAC_{120}$ into a compact lasso that could open into a relaxed conformation hypothetically activating catalysis.

## Sites of post-translational phosphorylations

*Cr*SBPase was reported to be phosphorylated from algae extracts at residues T112, S114, T118, S123, T310, S311, and T313 (*Wagner et al., 2006*; *Werth et al., 2017*; *McConnell et al., 2018*). Consistently with the proximity of these residues in two sectors of the sequence (*Figure 1—figure supplement 3*), the candidate modification sites are located in two contiguous regions in the three-dimensional structure (*Figure 1H*). T112, S114, T118, and S123 are exposed to the solvent of the amino-terminal

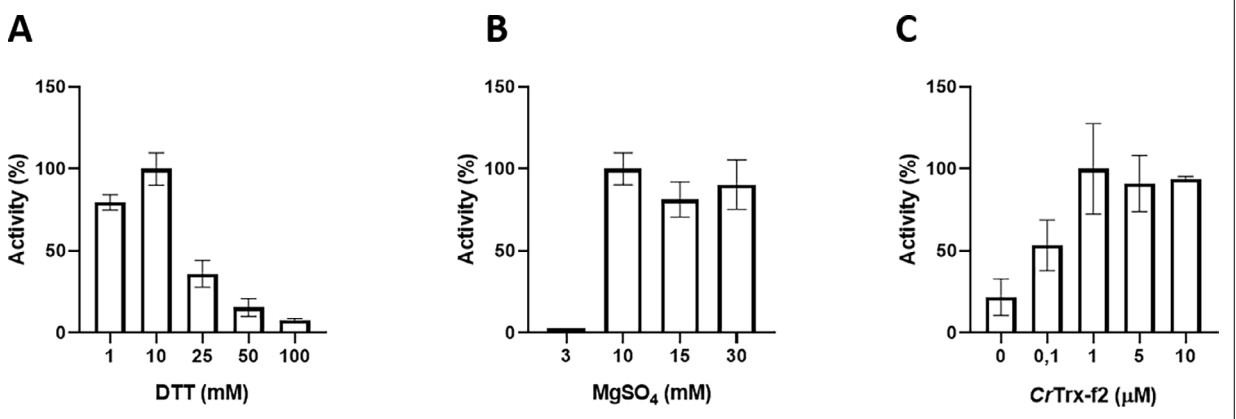

**Figure 2.** Functional characterization of *Chlamydomonas reinhardtii* SBPase (CrSBPase) in vitro. Reported enzymatic activity of *Cr*SBPase assayed for (**A**) reduction by reduced dithiothreitol (DTTred) and 10 mM of MgSO₄, (**B**) Magnesium sulfate (MgSO₄) and 10 mM DTTred, and (**C**) recombinant thioredoxin f2 from *Chlamydomonas reinhardtii* (*Cr*TRX-f2) and 10 mM MgSO₄, 1 mM DTT.

The online version of this article includes the following figure supplement(s) for figure 2:

**Figure supplement 1.** Regulatory properties of *Chlamydomonas reinhardtii* SBPase (CrSBPase) activity.

domain. The residues belong to or are located nearby the $A_{113}SCAGTAC_{120}$ motif where phosphorylations may cross-signal with redox modifications on C115 and C120. T310, S311, and T313 are located on the other edge of the enzyme, at the solvent exposed tip of the carboxy-terminal domain. Their co-localization at the $T_{310}SPT_{313}$ motif likely facilitates a coordinated phosphorylation of the three residues by the same kinase. Because both groups of potential phospho-sites are located at 17–18 Å from the catalytic pocket (W401 taken as a reference point), there is no straightforward mechanism by which such modifications would exert a control over *Cr*SBPase activity. Analysis of phospho-mimicking mutants along with the identification of the actual kinases/phosphatases couples that control *Cr*SBPase phosphorylation state would allow to comprehend the role of this putative regulatory mechanism.

## *Cr*SBPase enzymatic activity

Recombinant *Cr*SBPase was assayed for its capacity to catalyze FBP hydrolysis into F6P (*Gütle et al., 2016*), a proxy for the hydrolysis of SBP into S7P that we could not test by lack of available substrate and kinetic reporter method. Previous studies demonstrated that plant SBPase requires the cofactor $Mg^{2+}$ and a chemical reductant (DTTred) to be fully activated in vitro, reaching a $K_M$ value of 0.05 mM and a $k_{cat}$ of 81 sec$^{-1}$ in optimal conditions (*Anderson, 1974*; *Woodrow and Walker, 1980*; *Cadet and Meunier, 1988*). In agreement with this, we found a total absence of activity in the untreated *Cr*SBPase (data not shown), whereas pre-incubation of the enzyme with DTTred and $Mg^{2+}$ resulted in a strong activation, e.g., a detectable activity at DTT ≥1 mM (*Figure 2A*) and a stark increase in activity at MgSO₄ ≥10 mM (*Figure 2B*). By testing different concentrations of MgSO₄ (3, 10, 15, and 20 mM) and DTTred (1, 10, 25, 50, and 100 mM), we established that 10 mM of DTT and 10 mM MgSO₄ are the optimal conditions to attain the maximum specific activity (*Figure 2A and B*). Unexpectedly, DTTred concentrations above 10 mM had a dose-dependent inhibitory effect, causing a drastic drop in activity that probably resulted from structural effects leading to denaturation of the enzyme (*Figure 2A*, DTT ≥25 mM). In contrast, we observed no such inhibitory effect in the presence of magnesium at concentrations above 10 mM (*Figure 2B*). Under physiological conditions, the light-dependent redox regulation of plant SBPase is specifically exerted by f-type thioredoxins (*Gütle et al., 2016*), which are reduced by the ferredoxin-thioredoxin cascade (*Schürmann and Buchanan, 1975*). The chloroplast TRX-f2 from *Chlamydomonas reinhardtii* was thus tested and validated for its capacity to activate *Cr*SBPase by reduction, with maximal activation reached at TRX-f2 ≥1 µM and no apparent loss of activity at higher TRX-f2 concentrations (*Figure 2C*). The maximum SBPase reported activity, corresponding to 12.15±2.15 µmol(NADPH) min$^{-1}$ mg(SBPase)$^{-1}$, was obtained by pre-incubating the enzyme with 1 µM *Cr*TRX-f2 supplemented with 1 mM DTT and 10 mM MgSO₄, and maintained even at higher *Cr*TRX-f2 concentrations (i.e. 5 and 10 µM) (*Figure 2C*). *Cr*SBPase specific activity that we report is comparable to that reported for spinach SBPase with FBP as substrate (3.5 µmol min$^{-1}$ mg$^{-1}$

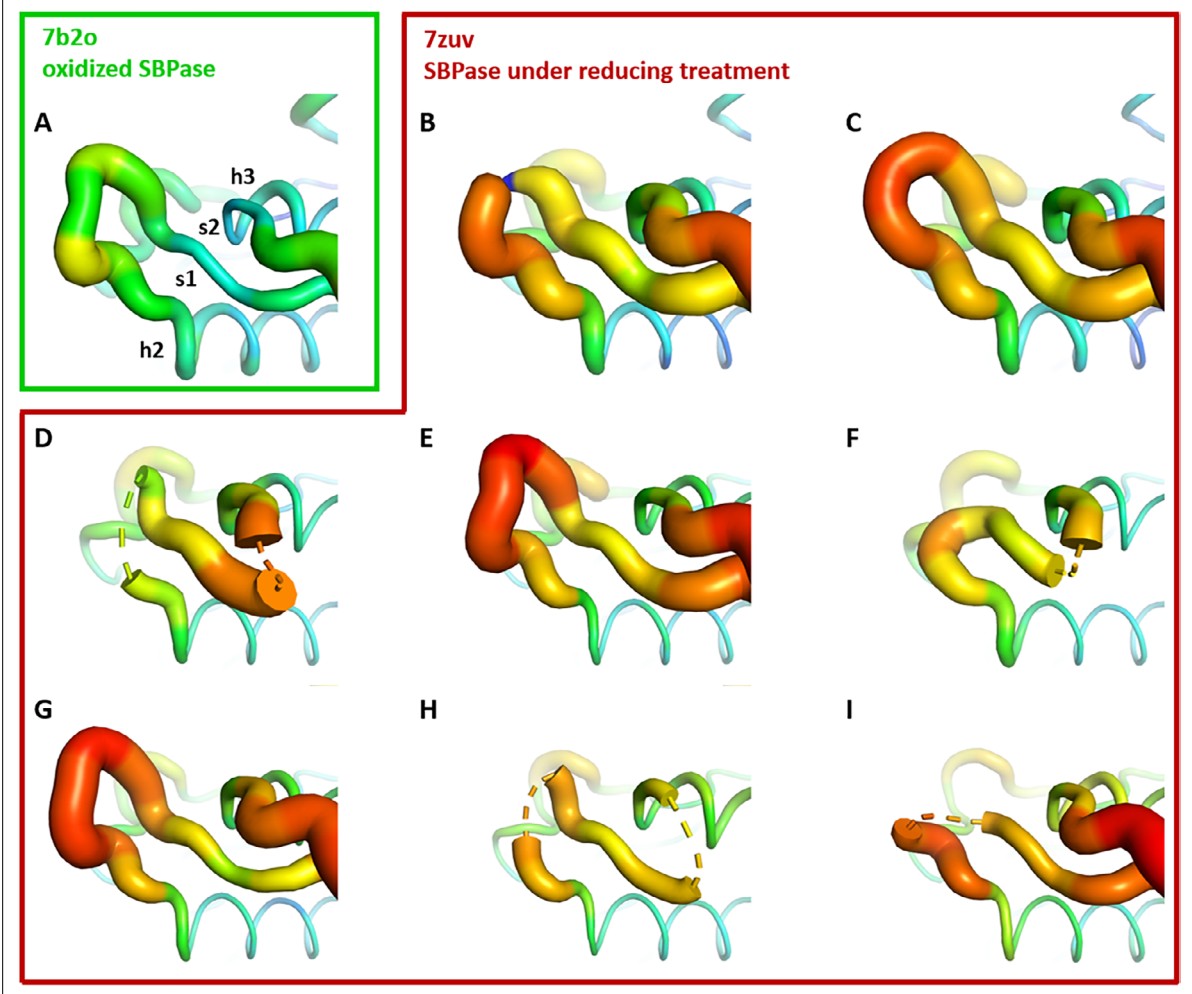

**Figure 3.** Crystallographic structures of *Chlamydomonas reinhardtii* SBPase (*Cr*SBPase) under reducing treatment: Local disorder of the $A_{113}SCAGTAC_{120}$ loop. Main chain was traced according to crystallographic b-factor, with large orange sections representing high b-factor values and thin blue sections representing low b-factors. (**A-I**). Aligned structures of *Cr*SBPase protomers without redox treatment (A, 7b2o chain A) or in the presence of 10 mM TCEP reducing agent (B-I, 7zuv chains A-H).

The online version of this article includes the following figure supplement(s) for figure 3:

**Figure supplement 1.** Crystallographic structures of *Chlamydomonas reinhardtii* SBPase (*Cr*SBPase) under reducing treatment: Local main chain rearrangement and variety of cysteine states of the $A_{113}SCAGTAC_{120}$ loop.

*Seuter et al., 2002*) and to that reported for another recombinant Chlamydomonas SBPase with SBP as substrate (7.15±0.61 µmol min⁻¹ mg⁻¹ *Tamoi et al., 2005*). We hence conclude that our assay faithfully recapitulate the level of *Cr*SBPase hydrolase function in its reduced state.

## Redox controlled dynamics of *Cr*SBPase

*Cr*SBPase crystallized under reducing treatment (10 mM TCEP). Among the eight subunits in the asymmetric unit, C115-C120 disulfide is absent in two chains (subunits C colored in magenta and subunits E colored in salmon, *Figure 3D and F*), while the other six subunits still have the C115-C120 disulfide bridge (*Figure 3B, C, E,G–I*). All other cysteines are unaffected by the reducing treatment, consistently with their thiol form in crystal structure of oxidized state. These six subunits present mildly variable conformations of the $A_{113}SCAGTAC_{120}$ lasso conformation similar to the untreated oxidized structure of *Cr*SBPase (*Figure 3A*). Reduction of the C115-C120 disulfide increases local disorder, and the main chain electron density became uninterpretable over a few residues for subunits C (residues 115–118, 124–127), and subunits E (residues 119–127) (*Figure 3—figure supplement 1*).

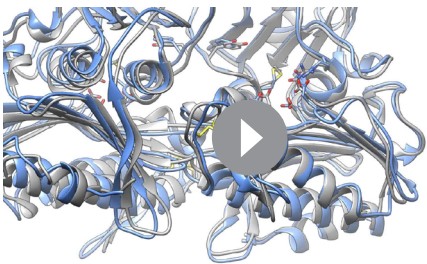

**Video 1.** Molecular dynamics simulation of *Chlamydomonas reinhardtii* SBPase (*Cr*SBPase) 7B2O chain A upon C115-120 disulfide bridge reduction. https://elifesciences.org/articles/87196/figures#video1

Molecular dynamics (MD) simulations of the reduced structure were performed by starting from a dimer of the crystallized oxidized form and forcing C115 and C120 residues to be reduced in the thiol state (-SH). We then ran two independent replicas of 20 μs of MD (*Videos 1 and 2*). In addition, we also ran two independent replicas of 2 μs of MD in the oxidized form. In one of the replicas of the reduced form (MD1), the protein structure barely changed as observed in the SG(C115)/SG(C120) distance that plateaued at ~7 Å, as well as in the RMSD with respect to the starting conformation that stayed around 2 Å (see *Figure 4—figure supplements 1–2* for plots and 12 for images of the structures). However, in the other replica of the reduced form (MD2), significant changes of conformation appeared: the SG(C115)/SG(C120) distance ended up at ~10 Å with peaks at 20 Å and the RMSD values ended up at 3.7 and 2.8 Å (*Figure 4—figure supplements 1–2*). The difference between MD1 and MD2 is stronger when one focuses on the RMSD of the mobile motif (residues 112–131, *Figure 4—figure supplement 3*). We observed that after 17 μs of MD all chains adopted an equilibrated conformation and then fluctuated around that conformation (*Figure 4—figure supplement 3*); representative structures extracted from MD were thus obtained from a clustering on the time frame (or block) 17–20 μs.

The two chains A and B from MD2 adopt a different final conformation (see *Figure 4A* for the overlap with the crystallized oxidized structure and *Figure 4B* for the overlap with the crystallized reduced structure), and we cannot directly conclude on which one is the most stable. Moreover, the residues 115–128 of each chain from MD1/2 do not overlap well with the crystallized reduced structure which means that neither of the two chains conformations from MD simulations has converged towards the crystallized state. However, we can observe in the reduced crystallized structure that residues 134–148 form an α-helix which is also present in chain B from MD2 whereas this structure is not kept in chain A from MD2 (see green chain on the right of *Figure 4A–B*). In addition, mobile motif from chain B seems to open towards the solvent in a similar way with what was observed for FBPase (*Chiadmi et al., 1999*). Thus, we conclude that chain B from MD2 is more representative of the true reduced conformation than chain A from MD2. If we compare the mobility of chain B from MD2 in the reduced state with one chain in the oxidized state (also chain B from MD2), we can see that the mobile motif is overall more flexible in the reduced state, which is especially true for residues around C120 (residues 118–123, *Figure 4—figure supplement 4*). We then analyzed the RMSD of residues 112–131 from chain B during MD2 (*Figure 4—figure supplement 5*): residues 112, 130, and 131 barely move which is expected since they belong or are directly connected to the two α-helixes flanking the mobile motif. At ~6 μs of the trajectory almost all other residues (113–129 residues with the sole exception of C120) display an abrupt change of conformation, with a different magnitude though: starting from ~2 Å, the RMSD jumped to 3 Å for some residues (residue 129) and to 15 Å for other residues (residues 125, 126) (*Figure 4—figure supplements 2–3–4*). At ~13.5 μs, another change occurs which involves C120 but not all the other residues. Thus, we conclude that the opening of the mobile motif happens in a concerted way and not sequentially residue per residue.

## Oligomeric states of *Cr*SBPase

Previous studies reported a homodimeric state for SBPase from the moss *Physcomitrella patens*, deduced from the quaternary structure of recombinant protein in the asymmetric unit of the crystal (*Gütle et al., 2016*) and in agreement with both

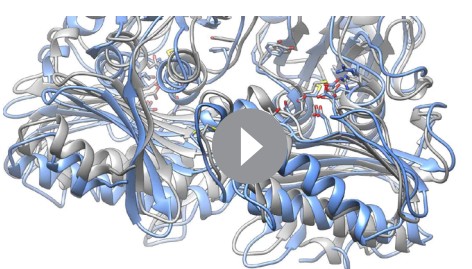

**Video 2.** Molecular dynamics simulation of *Chlamydomonas reinhardtii* SBPase (*Cr*SBPase) 7B2O chain B upon C115-120 bridge reduction. https://elifesciences.org/articles/87196/figures#video2

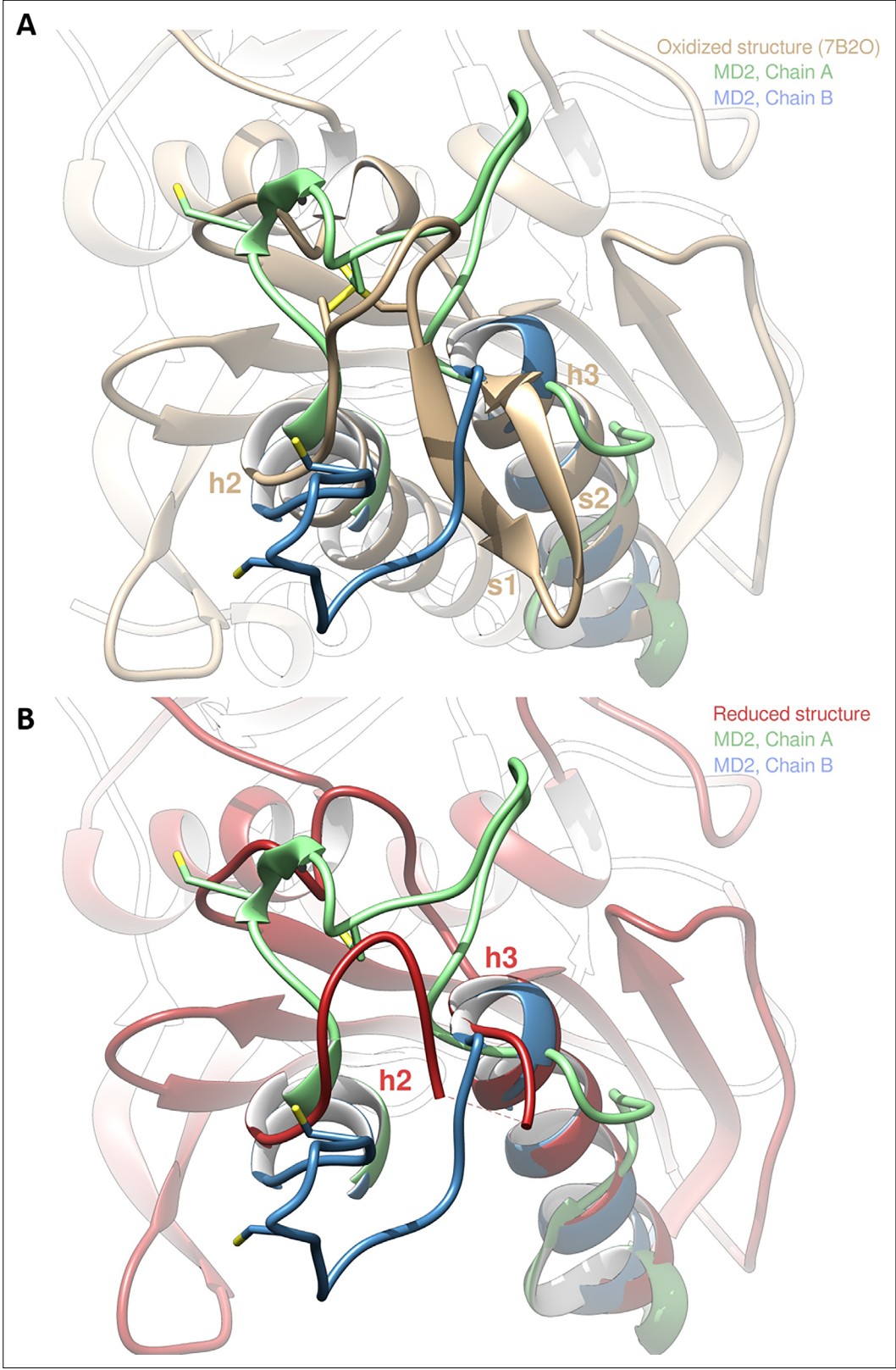

**Figure 4.** Molecular dynamics simulation of *Chlamydomonas reinhardtii* SBPase (*Cr*SBPase) after reduction.
(**A**) Overlap of the crystallographic structure of oxidized *Cr*SBPase and representative structures of equilibrated reduced *Cr*SBPase during molecular dynamics simulation 2 (MD2). For structures extracted from MD, only residues 109–148 are displayed since most of the other residues closely overlap those of the crystallographic structure.

*Figure 4 continued on next page*

*Figure 4 continued*

(**B**) Overlap of the crystallographic structure of reduced *Cr*SBPase and representative structures of equilibrated reduced *Cr*SBPase during MD2. For structures extracted from MD, only residues 109–148 are displayed.

The online version of this article includes the following figure supplement(s) for figure 4:

**Figure supplement 1.** Computational detail of molecular dynamic simulation: SG(Cys115)/SG(Cys120) distances along the molecular dynamics (MD) trajectories for each chain (left: oxidized state, right: reduced state).

**Figure supplement 2.** Computational detail of molecular dynamic simulation: RMSD of main chain along the molecular dynamics (MD) trajectories for each chain (left: oxidized state, right: reduced state).

**Figure supplement 3.** Computational detail of molecular dynamic simulation.

**Figure supplement 4.** Computational detail of molecular dynamic simulation: RMSF per residue for each chain (top row: full sequence, bottom row: zoom on residues 110–170; left column: oxidized state, middle column: reduced state, right column: comparison of one chain from each state).

**Figure supplement 5.** Computational detail of molecular dynamic simulation: RMSD of main chain of each residue from the mobile motif along the trajectory of chain B from MD2.

**Figure supplement 6.** Structures of *Chlamydomonas reinhardtii* SBPase (*Cr*SBPase) retrieved from molecular dynamics simulations 1 and 2.

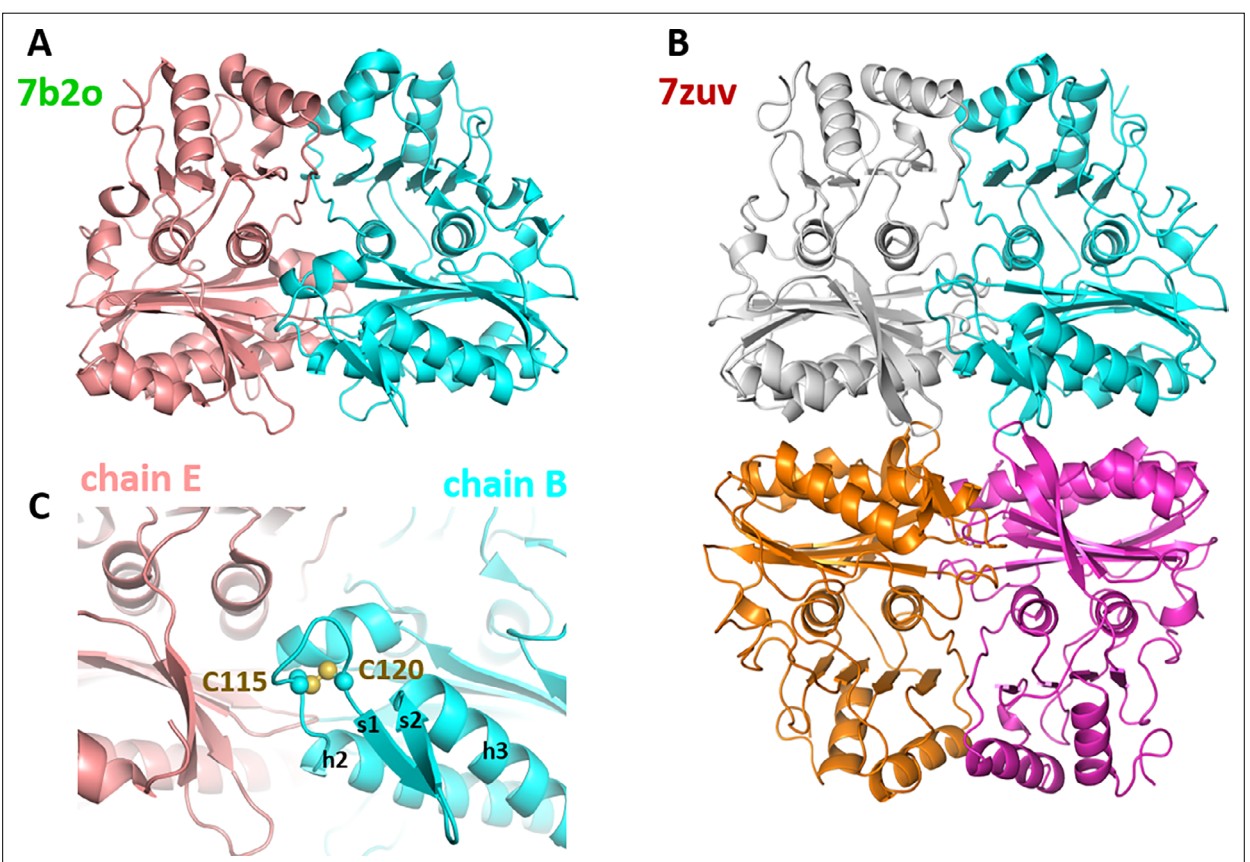

**Figure 5.** Oligomeric state of *Chlamydomonas reinhardtii* SBPase (*Cr*SBPase) in the crystal. (**A**) Asymmetric unit dimer of untreated *Cr*SBPase (7b2o). Chains are represented in cartoon and colored cyan (chain B) and salmon (chain E). (**B**) Asymmetric unit homotetramer under reducing treatment (7zuv). Chains A (cyan), C (magenta), F (white), and H (orange) belong to the same asymmetric unit. (**C**) Close-up view on figure A homodimer interface. Loop 113-ASCAGTAC-120 from chain B (in cyan) is in 5 Å distance of neighboring chain E (in salmon). C115 and C120 are bonded by a disulfide bridge.

The online version of this article includes the following figure supplement(s) for figure 5:

**Figure supplement 1.** Structure, oligomerisation and sequence variations according to X-ray crystallography and AlphaFold predictions.

our crystal structures and AF2/3 predictions detailed thereafter (*Figure 5*; *Figure 5—figure supplement 1*).

Size-exclusion chromatography (SEC) profile of *Cr*SBPase is polydisperse with at least three distinct oligomeric states (*Figure 6A*). Notably, we observed that the protein elutes as a main peak of 79 kDa, close to the mass of a homodimer (theoretical mass from monomer sequence: 36,176.7 Da) and consistently with the crystallized state of moss enzyme. Thorough inspection of the elution profile of *Cr*SBPase revealed a peak broadened by a later-eluting shoulder, with the apparent molecular mass of the tailing species at 48 kDa suggesting the presence of a monomeric state of *Cr*SBPase in vitro. A third *Cr*SBPase species elutes at an apparent molecular mass of 145 kDa, that fits with a homotetramer. According to the absorbance of each peak and considering that they represent all states of purified and oxidized *Cr*SBPase in solution, we estimate a repartition percentage of species of 10:53:37 for the tetramer:dimer:monomer mixture (total = 100%). Re-injection of the dimer peak fraction over size-exclusion chromatography yielded a similar mixed profile, supporting a dynamic equilibrium over the time of the experiment (>2 hr) (data not shown).

*Cr*SBPase was crystallized in space groups $P2_12_12$ or $P12_11$ with respectively 6 and 8 protomers in the asymmetric unit. Proteins Interfaces Structures and Assemblies (PISA) analysis reveals that protomers of both crystals pack as homodimers with extensive subunit interface of buried area 4170 Å² in a similar manner (*Figure 5A*; *Krissinel and Henrick, 2007*). In the asymmetric unit of *Cr*SBPase crystallized in reducing conditions (7zuv) or within crystallographic neighboring units of *Cr*SBPase in oxidized state (7b2o), *Cr*SBPase further packs into a homotetramer, a dimer of dimers (*Figure 5B*). The apparent molecular mass of SBPase extracted from *Chlamydomonas* cultures cultivated in TAP medium under 40 µE.m⁻².s⁻¹ illumination is comprised between 130 and 40 kDa with the homodimer representing the predominant state as supported by immunoblot analysis (*Figure 6B*).

To further investigate the oligomeric dynamics of CrSBPase, we performed small-angle X-ray scattering (SAXS) analysis of purified (untreated) or reduced enzyme in vitro which reveals an increase in the protein radius of gyration from 39 to 62 Å upon reduction (*Figure 6—figure supplement 1*). This observation implies a change in the oligomeric state of *Cr*SBPase correlating with the redox state. As a matter of fact, purified *Cr*SBPase tends to precipitate into amorphous aggregates that can be turned into droplets of liquid-liquid phase separation by treatment with 10 mM TCEP (*Figure 6—figure supplement 2*). Since the LBH redox module is positioned at the homodimer interface (*Figure 5C*), we postulated that the reduction of the C115-C120 disulfide bond and the subsequent local conformation changes depicted by MD affect the oligomer equilibrium of the enzyme. As a matter of fact, we observed that protein mutants C115S or C120S, which are unable to form the C115-C120 disulfide bond, elute in a different oligomer ratio than wild-type CrSBPase (*Figure 6A*). Indeed, we observed a relative accumulation of monomer and tetramer species at the expense of the dimer state in both mutants, as follows with tetramer:dimer:monomer ratio normalized on a scale of 100: WT=9:48:43; mutant C115S=42:25:33; mutant C120S=29:17:54. Size-exclusion chromatography suggests that C115-C120 is involved in the stabilization of the dimer. In order to test the hypothetical relevance of this disulfide bridge over catalysis, we assayed *Cr*SBPase-mC115S and *Cr*SBPase-m120S in the conditions described for wild-type enzyme in results section « *Cr*SBPase enzymatic activity » above. Both Cys mutants behaved as wild-type enzyme, *ie*. they responded to DTT/Mg-dependent activation resembling that of the C115-C120 disulfide-containing protein. No activity was detected for native and untreated forms, while specific activities of C115S and C120S mutants upon treatment with DTT and MgSO₄ (10 mM for both compounds) were near the specific activity of WT-SBPase (respectively 96% and 107% to the reference, 13.3±2.5 and 14,9±0,9 µmol(NADPH) min⁻¹ mg(SBPase)⁻¹ for C115S and C120S mutants) (*Figure 2—figure supplement 1*). Considering the standard deviation of measured activities, there is no statistically significant difference between WT and mutants in our assays. Based on these data, we can thus conclude that C115-C120 disulfide bond is not the sole thiol-dependent regulatory mechanism involved in the modulation of *Cr*SBPase activity and that the redox dependency of *Cr*SBPase catalysis likely depends on one or several other cysteine pairs. Future studies are required to shed light on the identity of cysteine residues that contribute to *Cr*SBPase catalytic modulation, other than C115 and C120.

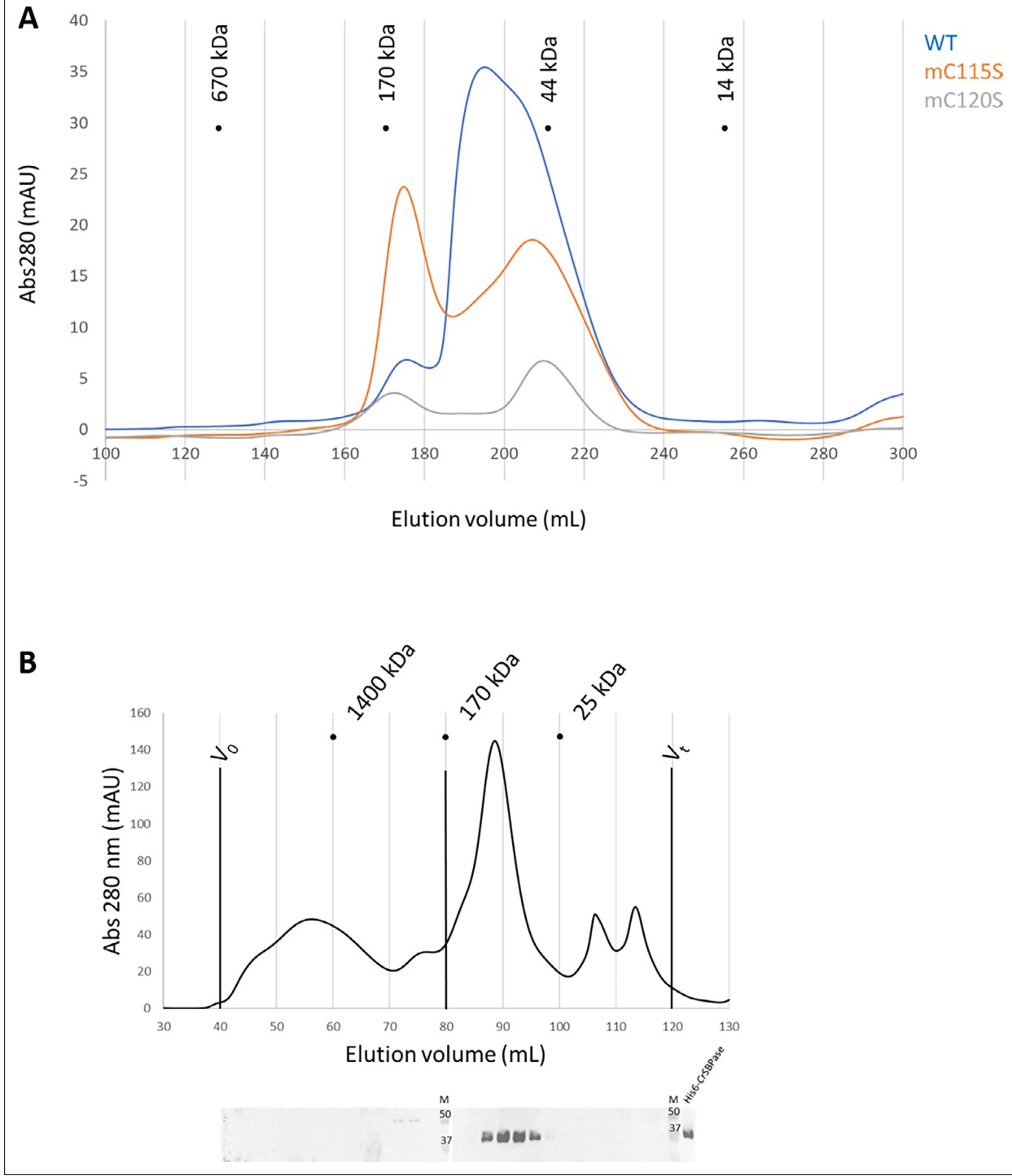

**Figure 6.** Oligomeric state of *Chlamydomonas reinhardtii* SBPase (*Cr*SBPase) in vitro. (**A**) Size-exclusion chromatograms of *Cr*SBPase wild-type (blue), mutant C115S (orange), mutant C120S (gray) on Superdex 200 26/600 GL column. (**B**) Size-exclusion fractionation of Chlamydomonas cell extracts. Chlamydomonas cell culture was harvested, lysed and the soluble fraction of the lysate was loaded on Superose6 16/600 size-exclusion column. Chromatography fractions were analyzed by western blot with anti-*Cr*SBPase primary antibodies. First membrane was loaded with fractions eluted from 40 to 80 mL. Second membrane was loaded with fractions eluted from 80 to 120 mL. M lane is loaded with molecular mass standards ladder. Recombinant *Cr*SBPase was loaded on the last lane to the right.

The online version of this article includes the following source data and figure supplement(s) for figure 6:

**Source data 1.** Original western blots shown in *Figure 6B*.

*Figure 6 continued on next page*

*Figure 6 continued*

**Source data 2.** Original western blots shown in *Figure 6B* (labelled).

**Figure supplement 1.** Size-exclusion chromatography coupled to small angle X-rays scattering (SEC-SAXS) of *Chlamydomonas reinhardtii* SBPase (*Cr*SBPase).

**Figure supplement 2.** Phase separation of reduced *Chlamydomonas reinhardtii* SBPase (*Cr*SBPase) precipitates.

## Discussion

We report the first crystal structure of an SBPase from the microalga *Chlamydomonas reinhardtii*, a model for the molecular and cell biology of plants. SBPase is a photosynthetic enzyme involved in the regeneration phase of the CBBC cycle proved to be target of multiple regulations that we map onto our structural model. The native folding of *Cr*SBPase is highly similar to that of SBPase from the moss *Physcomitrium patens* and to FBPase from chloroplast or cytoplasmic origins. Notwithstanding its propensity to form crystals, recombinant *Cr*SBPase encompasses a range of oligomeric states, dominated by a homodimeric form susceptible to exchange for monomers, tetramers, and higher-order assemblies. This oligomer equilibrium is dynamic and correlates with the presence of a disulfide bridge at the vicinity of the monomer-monomer interface, in the SBPase-specific LBH motif. We postulate that the reversible reduction/oxidation of the C115-C120 pair modulates the oligomerization of *Cr*SBPase, through the induction of a local disorder that we observed in the crystal structure of the protein under reducing conditions and that correlates with computations of molecular dynamics.

How this oligomeric exchange impacts the actual activity of the enzyme is an open question but should be aligned with previous reports on the allosteric character of the structurally similar FBPase enzyme which follows dimer/tetramer exchanges (*Barciszewski et al., 2016*) with a Hill cooperativity coefficient close to 2 representing positive cooperativity in the context of a homodimer (*Giudici-Orticoni et al., 1990*). Recent in vivo proteome mapping in *Chlamydomonas reinhardtii* localized SBPase in a punctate region surrounding the Rubisco pyrenoid nearby five other CBBC enzymes: phosphoglycerate kinase 1 (PGK), glyceraldehyde-3-phosphate dehydrogenases (GAP1, GAP3), fructose-bisphosphate aldolase (FBA3), ribulose-5-phosphate 3-epimerase (RPE1), and phosphoribulokinase (PRK) (*Wang et al., 2022*). Based on our structure-function study on recombinant SBPase, we can speculate that the altered redox state of SBPase may influence the movement to or from the liquid partition of the stroma, where SBPase is metabolically required to provide its contribution to CBBC reactions near other enzymes of the pathway.

## Materials and methods

Chemical reagents were purchased from Merck (Darmstadt, Germany), chromatography material was provided by Cytiva (Vélizy-Villacoublay, France), crystallization solutions and consumable were purchased from Qiagen (Hilden, Germany), Hampton Research (Aliso Viejo, CA USA), and SPT Labtech (Melbourn, UK).

### Cloning and mutagenesis of *Cr*SBPase expression plasmids

The sequence coding for full-length mature SBPase was amplified by reverse transcriptase polymerase chain reaction (RT-PCR) from *Chlamydomonas reinhardtii* total ribonucleic acids extract and inserted at sites 5'-NdeI and BamHI-3' into expression vector pET3c modified for the fusion of an amino-terminal hexa-histidine tag in the recombinant protein, yielding plasmid pSL-175-His$_6$-*Cr*SBPase-WT. PCR primers were 5'-TCTCGCCATATGGCCGTTCTGACCCAGGCC-3' (forward, NdeI site underlined) and 5'-CGGGTGGGATCCTTAGGCAGCCACCTTCTC-3' (reverse, BamHI site underlined). The encoded

**Table 1.** Primers used for point mutagenesis.

| 5'-forward C115S | CCGCACCGCCTCGAGCGCCGGTACCGCCTGCGTG |
| --- | --- |
| 3'-reverse C115S | CACGCAGGCGGTACCGGCGCTCGAGGCGGTGCGG |
| 5'-forward C120S | GCGCCGGTACCGCCAGCGTGAACAGCTTCGGCG |
| 3'-reverse C120S | CGCCGAAGCTGTTCACGCTGGCGGTACCGGCGC |

protein encompasses 331 amino acids, with a molar mass of 36,176.7 Da and a theoretical extinction coefficient of 24,722.5 $M^{-1}.cm^{-1}$. Unless otherwise stated, amino acid residues numbering throughout the manuscript is that of GenBank entry DAA79954.1 and Uniprot entries A8IRK4 and P46284. Point mutants of cysteines 115 and 120 singly substituted with serines were generated by site-directed PCR mutagenesis with primers listed in *Table 1*, producing plasmids pSL-181-His6-*Cr*SBPase-C115S, and pSL-182-His6-*Cr*SBPase-C120S.

## Recombinant *Cr*SBPase purification

Plasmids pSL-175,–181, and –182 were used to transform *Escherichia coli* BL21(DE3). The transformants were grown at 37 °C in 2YT supplemented with 150 µg/mL ampicillin. When cultures reached an $OD_{600}$ of ~0.6, T7-dependent *Cr*SBPase overexpression was induced by the addition of isopropyl-β-D-thiogalactopyranoside (IPTG) at 0.2 mM for 3 h.

Harvested cell pellets were resuspended in 20 mM Tris pH 7.9, 100 mM NaCl (buffer A) and lyzed by 1 s/1 s pulsed sonication for 3 min. The soluble fraction of the lysate was separated by 20 min centrifugation at 20,000 *g* and loaded on 2 mL Ni-NTA resin for His-tag mediated affinity chromatography. The resin was washed in four steps with 25 mL buffer A, 25 mL buffer A supplemented with 10, 20, and 30 mM imidazole, and *Cr*SBPase was eluted in three steps with 10 mL buffer A supplemented with 100, 200, and 300 mM imidazole. Pooled eluates containing *Cr*SBPase were loaded on HiLoad Superdex 200 26/600 size-exclusion chromatography column and eluted in buffer A. Protein purity was assessed by electrophoresis on 12% acrylamide gel under denaturing and reducing conditions (*Figure 1—figure supplement 1*). Peak fractions were assembled and concentrated by ultrafiltration on Amicon units of MWCO 30,000 Da cut-off. Final concentrations of purified proteins were in the 1–10 mg/ml range as measured by Nanodrop ND-2000 spectrophotometer using an $\varepsilon_{280}$ of 24,722.5 $M^{-1}.cm^{-1}$.

## Western blot

Polyclonal antibodies were generated by immunization of rabbits with pure recombinant *Cr*SBPase (Genecust, Boynes France). Analyzed fractions were separated by denaturing polyacrylamide electrophoresis and transferred to 0.2 µm nitrocellulose membrane for detection with primary antibody subsequently detected by secondary anti-rabbit antibodies coupled to horseradish peroxidase. Detection was done with commercial ECL peroxidase assay (GE Healthcare, Chicago IL USA) with a Chemidoc (Bio-Rad, Hercules CA USA).

## Crystallization and structure determination

Pure recombinant His6-*Cr*SBPase-WT concentrated at 3–6 mg/mL was tested for crystallization on screens JCSG I-IV in 200 nL sitting drops. Single crystals were obtained and optimized with mother liquor 0.1 M sodium HEPES pH = 7.5, 2% (v/v) polyethylene glycol 400, 2.0 M ammonium sulfate, flash-frozen in mother liquor supplemented with 25% glycerol and tested for X-ray diffraction. Complete X-ray diffraction datasets were collected at SOLEIL beamline Proxima-2A and allowed the determination of the crystal structure by model replacement with 5iz3, the structure of ortholog SBPase from the moss *Physcomitrium patens*. Model building in COOT and refinement in the PHENIX software suite yielded structure deposited under PDB ID 7b2o (*Emsley et al., 2010*; *Adams et al., 2011*; *Liebschner et al., 2019*) (PDB DOI: https://doi.org/10.2210/pdb7B2O/pdb) (*Table 2*). After we obtained our experimental structure, the European Bioinformatics Institute (Hinxton, UK) communicated high-accuracy prediction models computed by Deepmind AlphaFold 2 (*Jumper et al., 2021*), including a prediction of *Cr*SBPase structure (https://alphafold.ebi.ac.uk/entry/P46284) that matches chain A in 7b2o crystal structure with a root mean square deviation (RMSD) of 0.453 Å. Hence, X-ray crystallography and AlphaFold independently contribute to the proposed structural analysis.

In order to obtain the structure of *Cr*SBPase in a reduced state, crystals were grown in the presence of 10 mM of the reducing agent tris-(2-carboxyethyl)phosphine (TCEP) in condition JCSG IV E8/56 0.2 M lithium sulfate, 0.1 M Tris pH 7.0, and 1.0 M sodium/potassium tartrate. Crystals were cryoprotected in mother liquor supplemented with 25% ethylene glycol, flash-frozen, and tested for X-ray diffraction at SOLEIL beamline Proxima-2A. A complete dataset was collected that allowed the determination of the structure by molecular replacement with 7b2o as a template, resulting in structure 7zuv deposited in the protein data bank (PDB DOI: https://doi.org/10.2210/pdb7ZUV/pdb). Among

**Table 2.** Crystallographic data collections and models building statistics.

| | 7b2o (untreated) | 7zuv (partially reduced) |
|---|---|---|
| Wavelength (Å) | 0.9801 | 0.9801 |
| Resolution range (Å) | 46.62–3.095 (3.206–3.095) | 48.54–3.11 (3.221–3.11) |
| Space group | P 2$_1$ 2$_1$ 2 | P 1 2$_1$ 1 |
| Unit cell (Å, °) | 178.224 183.652 75.196 90 90 90 | 53.774 163.462 172.765 90 91.939 90 |
| Total reflections | 615155 (55989) | 376091 (32376) |
| Unique reflections | 45740 (4245) | 53456 (5213) |
| Multiplicity | 13.4 (12.9) | 7.0 (6.2) |
| Completeness (%) | 99.34 (94.20) | 99.65 (97.17) |
| Mean I/sigma (I) | 12.02 (1.72) | 8.54 (1.30) |
| Wilson B-factor (Å²) | 64.82 | 68.45 |
| R-merge | 0.495 (1.877) | 0.2492 (1.502) |
| R-meas | 0.5144 (1.954) | 0.2691 (1.64) |
| R-pim | 0.1386 (0.5343) | 0.1009 (0.6477) |
| CC1/2 | 0.983 (0.623) | 0.988 (0.479) |
| CC* | 0.996 (0.876) | 0.997 (0.805) |
| Reflections used in refinement | 45607 (4242) | 53417 (5193) |
| Reflections used for R-free | 1986 (183) | 1983 (191) |
| R-work | 0.1942 (0.3071) | 0.1963 (0.3239) |
| R-free | 0.2390 (0.3482) | 0.2328 (0.3740) |
| CC (work) | 0.944 (0.788) | 0.956 (0.726) |
| CC (free) | 0.912 (0.649) | 0.949 (0.596) |
| Number of non-hydrogen atoms | 14194 | 18714 |
| Macromolecules | 14190 | 18704 |
| Solvent | 4 | 10 |
| Protein residues | 1859 | 2451 |
| RMS (bonds) (Å) | 0.005 | 0.004 |
| RMS (angles) (°) | 0.75 | 0.73 |
| Ramachandran favored (%) | 94.26 | 94.80 |
| Ramachandran allowed (%) | 5.52 | 4.70 |
| Ramachandran outliers (%) | 0.22 | 0.50 |
| Rotamer outliers (%) | 0.13 | 0.55 |
| Clashscore | 4.88 | 6.60 |
| Average B-factor (Å²) | 67.00 | 68.41 |
| Macromolecules (Å²) | 67.01 | 68.40 |
| Solvent (Å²) | 49.59 | 94.37 |

Statistics for the highest-resolution shell are shown in parentheses.

the eight *Cr*SBPase copies of the asymmetric unit, two had their disulfides absent as a consequence of the TCEP treatment while the other six subunits were essentially identical to untreated structures in entry 7b2o (i.e. fully oxidized form).

## Enzymatic assays

The catalytic activity of 10–200 nM pure recombinant *Cr*SBPase was spectrophotometrically assayed in buffer A by coupling the hydrolysis of 1 mM FBP into F6P to the reduction of 0.5 mM NADP$^+$ into NADPH through isomerization of F6P into glucose-6-phosphate (G6P) by 0.5 U.mL$^{-1}$ phosphoglucose isomerase (PGI) and the oxidation of G6P into 6-phosphogluconolactone (6 PGL) by 0.1–0.5 U.mL$^{-1}$ glucose-6-phosphate dehydrogenase (G6PDH). Reporter assay is coupled in a molar ratio FBP:F6P:G6P:6 PGL of 1:1:1:1 to the oxidation of one molar equivalent of NADP$^+$ into NADPH that we recorded over 20 min by measuring the absorbance at 340 nm using a UVIKON spectrophotometer. Pre-treatments with 0–100 mM reduced dithiothreitol (DTTred), 0–30 mM MgSO$_4$, and 0–15 μM recombinant *Cr*TRX-f2 supplemented with 1 mM DTTred (*Lemaire et al., 2018*) were carried out for 30 min at room temperature before measuring enzyme activity as described above. Experiments were conducted in technical triplicates (WT-*Cr*SBPase) or duplicates (mutants).

## Titration of accessible cysteine thiols

Quantification of free reactive cysteines on recombinant CrSBPase was performed with 5,5'-dithio-bis-(2-nitrobenzoic acid) (DTNB, Ellman's reagent) mixed with the purified recombinant protein in a 200:1 molar ratio in a 1 mL reaction volume containing 100 mM Tris-HCl, pH 7.9 as described previously (*Meloni et al., 2024*). Reaction of DTNB with solvent-exposed cysteine thiols produces 2-nitro-5-thiobenzoate (TNB$^-$) that we quantified by monitoring the absorbance at 412 nm and a molar extinction coefficient of TNB$^-$ of 14.15 mM$^{-1}$.cm$^{-1}$. We calculated the number of free reactive cysteines per *Cr*SBPase monomer by dividing the molar concentration of TNB$^-$ by that of the protein.

## Molecular dynamics

Molecular dynamics (MD) simulations were performed with the Gromacs package (v2021.4). A dimer of CrSBPase was solvated in a rhombic dodecahedron box with at least 8 Å between the protein atoms and the box edges, which corresponds to ~64,000 atoms in total (~18,000 water molecules) and a triclinic box of 97 × 97 × 68 Å$^3$. The box was neutralized with 18 sodium atoms to make it neutral. The starting conformation was the crystallized structure of oxidized CrSBPase, and we protonated C115 and C120 to manually create the reduced state. The protein was described with the Amber14SB force field and water with the TIP3P model. Non-bonding interactions were described with standard parameters: van der Waals interactions had a cut-off of 8 Å, and electrostatic interactions were computed with PME with default parameters with a separation between spaces at 8 Å. Bonds containing a hydrogen atom were restrained with the LINCS algorithm with default parameters. After an energy minimization, the system was equilibrated under the NPT conditions with a δt=1 fs timestep, the velocity-rescale thermostat, and the Berendsen barostat. Velocities were generated at 100 K and the system was heated up to 300 K in 400 ps before performing 100 ps at 300 K. We then ran simulations of 20 μs under the NPT conditions with a δt=2 fs timestep, the velocity-rescale thermostat and the Parrinello-Rahman barostat.

## Cultivation of Chlamydomonas for native SBPase analysis

*Chlamydomonas reinhardtii* strain D66 was grown in 50 mL Tris-acetate-phosphate (TAP) medium under 40 μE.m$^{-2}$.s$^{-1}$ continuous light until density reached 5 × 10$^4$ cell.mL$^{-1}$, harvested by 10 min centrifugation at 3,000 g. Cell pellet was resuspended in 100 mM Tris-HCl, pH 7.9, 100 mM NaCl, 5 mM L-cystéine, 500 μM NADH, and 1 mM EDTA (buffer L) and lysed by three freeze-thaw cycles in liquid nitrogen and water bath. Soluble fraction was separated by 10 min centrifugation at 30,000 *g* and loaded for size-exclusion chromatography on a calibrated Superose 6 16/600 column equilibrated in buffer L. Aliquots of 500 μL elution fractions were analyzed by western blot with primary antibodies raised in rabbits against pure recombinant *Cr*SBPase (Covalab, Bron, France) and secondary anti-rabbit IgG coupled to peroxidase (Sigma-Aldrich reference A9169, Saint Louis, USA) revealed by ECL Prime (GE Healthcare).

## Small-angle X-ray scattering

Pure recombinant *Cr*SBPase was injected on Agilent HPLC BioSEC-3 300 column in buffer A, in line with the X-ray beam capillary at SOLEIL beamline SWING. *Cr*SBPase was untreated or pre-treated with 10 mM of reducing agent dithiothreitol (DTTred). Diffusion curves were analyzed with ATSAS software to determine the molecular mass and oligomeric state of the protein.

## Materials availability statement

Newly created materials can be accessed without any particular restriction upon request to corresponding author.

## Acknowledgements

We acknowledge the Institut de Biologie Physico-Chimique (CNRS, FR 505) for access to the crystallization facility. We acknowledge SOLEIL for provision of synchrotron radiation facilities at beamlines SWING, PROXIMA-1, and PROXIMA-2A. We thank Guillaume Q Robert for his contribution to protein purification, crystallization under reducing treatment, and model building. Plasmids pSL-175,–181, and –182 were cloned by Dr. Laure Michelet. This work was funded by the grant CALVINTERACT from the Agence Nationale de la Recherche (ANR-19-CE11-0009). Martina Santoni received funding from the University of Bologna and Sorbonne University exchange program ERASMUS+. This work was performed using HPC resources from GENCI–IDRIS (Grant 2021–077156). The authors express their gratitude to the financial support of the CNRS-INSB through a grant «Accès aux infrastructures nationales».

## Additional information

### Funding

| Funder | Grant reference number | Author |
| --- | --- | --- |
| Centre National de la Recherche Scientifique | INSB-D-2022-135 | Julien Henri |
| GENCI-IDRIS | 2021-077156 | Nicolas Chéron |
| Agence Nationale de la Recherche | ANR-19-CE11-0009 | Julien Henri |
| University of Bologna | ERASMUS+ | Martina Santoni |
| Sorbonne University | ERASMUS+ | Martina Santoni |

The funders had no role in study design, data collection and interpretation, or the decision to submit the work for publication.

### Author contributions

Théo Le Moigne, Validation, Investigation, Methodology, Writing - original draft; Martina Santoni, Investigation, Methodology, Writing - original draft; Lucile Jomat, Investigation; Stéphane D Lemaire, Conceptualization, Supervision, Writing - original draft; Mirko Zaffagnini, Conceptualization, Supervision, Investigation, Methodology, Writing - original draft, Writing - review and editing; Nicolas Chéron, Conceptualization, Validation, Investigation, Visualization, Methodology, Writing - original draft, Writing - review and editing; Julien Henri, Conceptualization, Supervision, Funding acquisition, Validation, Investigation, Visualization, Methodology, Writing - original draft, Project administration, Writing - review and editing

### Author ORCIDs

Julien Henri ORCID https://orcid.org/0000-0003-0772-8881

Joint public review: https://doi.org/10.7554/eLife.87196.4.sa1
Author response https://doi.org/10.7554/eLife.87196.4.sa2

## Additional files

### Supplementary files
MDAR checklist

### Data availability
Diffraction data have been deposited in PDB under the accession codes 7B2O and 7ZUV.

The following datasets were generated:

| Author(s) | Year | Dataset title | Dataset URL | Database and Identifier |
|---|---|---|---|---|
| Le Moigne T, Lemaire SD, Henri J | 2021 | 7B2O Crystal structure of *Chlamydomonas reinhardtii* chloroplastic sedoheptulose-1,7-bisphosphatase | https://doi.org/10.2210/pdb7B2O/pdb | Worldwide Protein Data Bank, 10.2210/pdb7B2O/pdb |
| Le Moigne T, Robert GQ, Lemaire SD, Henri J | 2023 | 7ZUV Crystal structure of *Chlamydomonas reinhardtii* chloroplastic sedoheptulose-1,7-bisphosphatase in reducing conditions | https://doi.org/10.2210/pdb7ZUV/pdb | Worldwide Protein Data Bank, 10.2210/pdb7ZUV/pdb |

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
