## [Editor Report · eLife Assessment]

This study provides a **valuable** structural analysis of the Sedoheptulose-1,7-Bisphosphatase (SBPase) from *Chlamydomonas reinhardtii*. The data presented are **solid** and based on X-ray structures of the CrSBPase in an oxidized and reduced state, the authors identify a disulfide bond in close proximity to the dimer interface. They show that the redox-state of the CrSBPase impacts its oligomeric state and might also influence the activity of the protein.

---

## [Referee Report · Joint public review]

The central theme of the manuscript is the structure of SBPase - an enzyme central to the photosynthetic Calvin-Benson-Bassham cycle. The authors claim that the structure is first of its kind from a chlorophyte *Chlamydomonas reinhardtii*, a model unicellular green microalga. The authors use a number of methods like protein expression, purification, enzymatic assays, SAXS, molecular dynamics simulations and xray crystallography to resolve a 3.09 A crystal structure of the oxidized and partially reduced state. The results are supported by the claims made in the manuscript. While the structure is the first from a chlorophyte, it is not unique. Several structures of SBPase are available and a comparison has been made between the structure reported here and others that have been previously published.

---

## [Author Response]

The following is the authors’ response to the previous reviews.

**Recommendations for the authors:**

**Reviewer #1:**
The authors have thoroughly changed the manuscript and addressed most of my concerns. I appreciate adding the activity assays of the C115/120S mutants, however, I suggest that the authors embed and also discuss these data more clearly. It also escaped my attention earlier that the positioning of the disulfide bond is 117-122 in the deposited PDBs instead of 115-120. The authors should carefully check which positioning is correct here.

We thank reviewer #1 for his or her careful assessment of our revised manuscript. As suggested, we detailed the results section “*Cr*SBPase enzymatic activity” with additional numerical values, and discussed more clearly the comparisons of results for activity assays of mutants C115S and C120S in the section “Oligomeric states of *Cr*SBPase”. Residues numbering was carefully proof-checked throughout the manuscript for correctness and homogeneity. C115 and C120 are numbered according to best databases consensus, *ie.* GenBank and Uniprot, and may differ from one database to another (including PDB) due to varying numbering rules. We clarified the chosen nomenclature in methods section “Cloning and mutagenesis of *Cr*SBPase expression plasmids”.

Line 246-250: I think it is evident that the two SBPase structures superpose well given the sequence identity of more than 70%. However, it would be great to include a superposition of the two structures in Figure 1, especially with regard to the region harboring C115 and C120.

We added a panel showing superimposition of *Cr*SBPase 7b2o and *Pp*SBPase 5iz3 and made a close-up view around the region C115-C120 in supplementary figure 5. Given the density in information of figure 1 we prefer not to add additional images on it. Supplementary figure 5 was initially intended to illustrate sequence conservation/variation among homologs, thus fitting with the objective to compare past and present XRC results.

Line 255-266: I am again missing a panel in Figure 1 here, e.g. a side-by-side view of Xray vs AF2/3 structure.

We added another panel in supplementary figure 5 to visually compare side-by-side SBPase crystallographic structure 7b2o and our AF3 model. Again, for the sake of clarity we prefer not to overload figure 1 with additional panels. This will also enable thorough comparison of past XRC of *Pp*SBPase, present XRC of *Cr*SBPase, and various AF models (see below, oligomer comparisons).

Line 261-266: Did the authors predict dimers and tetramers using AF3? What are the confidence metrics in this case? Do the authors see differences to the monomer prediction in case a multimer is confidently predicted?

We modeled dimers and tetramers using AF3 and added them on supplementary figure 5 side by side with protomer of XRC model 7b2o and with monomer predicted by AF3. Color code for supplementary figure 5 panels F-H is according to AF standard representation of plDDT. Confidence metrics per residue correspond to very high reliability (navy blue) or, locally, confident prediction (cyan) and overall prediction scores range from pTM=0.85-0.91, a high-quality prediction. Interface prediction score is high for both dimer (ipTM=0.9) and tetramer (ipTM=0.82). We reported these data in supplementary figure 5 and corresponding updated legend. XRC and AF models all align with RMSD<0.5 Å, indicating a globally unchanged structure of the protomer in the various methods and oligomeric states.

Line 441: How does the oligomeric equilibrium change in C115/120S mutants? This information should be added for the mutants. Besides, the mAU units in Fig. 6 could be normalized to allow an easier comparison between the chromatograms of wt and mutants.

Change in oligomeric equilibrium is assessed by size-exclusion chromatography of WT and mutants C115S, C120S as reported in figure 6A. We made quantitative estimation of WT, and C115S and C120S mutants equilibrium by comparing maximal peak intensity and added this information in the text. Briefly, the oligomer ratio on a scale of 100 is 9:48:43 for WT, 42:25:33 for mutant C115S, and 29:17:54 for mutant C120S (ratio expressed as tetramer:dimer:monomer). We prefer not to normalize values of absorbance, but rather keep the actual measurement of absorbance at 280 nm on the chromatogram of figure 6, for the sake of consistency with the added text and for a more transparent report of the experiment.

Line 447: WT activity is 12.15+-2.15 and both mutants have a higher activity. The authors should check if their values (96% and 107%) are correct. Besides, did the authors check if the increase in C120S is statistically significant? My impression is that both mutants have a higher activity than the wildtype, in both correlating with increased fractions of the tetramer. This would also make sense, as the corresponding region is part of the tetramer interface in the crystal packing.

The reported activity values were checked for correctness. Wild-type SBPase specific activity at 12.5 ±2.15 µmol(NADPH) min^-1^ mg(SBPase)^-1^ was obtained by pre-incubating the enzyme with 1 µM *Cr*TRXf2 supplemented with 1 mM DTT and 10 mM Mg^2+^, while the results of supplementary figure 14 reporting the comparison of activation of WT and mutants, with a variation of 107 or 96 %, were obtained with a slightly different protocol for pre-incubation of the enzyme with 10 mM DTT and 10 mM Mg^2+^. Please note that whether WT enzyme was assayed in 10 mM DTT 10 mM Mg or in 1 µM TRX 1 mM DTT 10 mM Mg, its specific activity appears equal within experimental error. Both mutants have nearly the same activity than the WT in the assay reported in supplementary figure 14: we fully agree that 107% (and 96%) variation is indeed not significant considering the uncertainty of the measurement (see error bars representing standard deviations of the mean in supplementary figure 14). We added this important information in the text. Even though both mutations stabilize the most active tetramer in untreated recombinant protein, we think that after reducting treatment both WT and mutants all reach the same maximal activity because they all form an equivalent proportion of the active tetramer versus alternative oligomeric states. We furhter interprete this piece of data as a decoupling of reduction and catalysis: in physiological conditions we assume that SBPase would initiate activation upon the reduction of disulfide bridges, including but not limited to C115-C120 that restricts the entry into fully active tetramer, at which point SBPase in reduced form reaches maximal activity until another post-translational signal eventually changes its conformation and oligomerisation.

We thank again reviewer 1 for his or her assessment and valuable suggestions.